# The Role of NLRP3 Inflammasome in Obesity and PCOS—A Systematic Review and Meta-Analysis

**DOI:** 10.3390/ijms241310976

**Published:** 2023-07-01

**Authors:** Salih Atalah Alenezi, Raheela Khan, Lindsay Snell, Shaimaa Aboeldalyl, Saad Amer

**Affiliations:** 1Division of Translational Medical Sciences, School of Medicine, Royal Derby Hospital Centre, University of Nottingham, Derby DE22 3DT, UK; salih.alenezi@nottingham.ac.uk (S.A.A.); raheela.khan@nottingham.ac.uk (R.K.); 2Prince Mohammed Bin Abdulaziz Medical City, Ministry of Health, Riyadh 14214, Saudi Arabia; 3University Hospitals of Derby and Burton NHS Foundation Trust, Library & Knowledge Service, Derby DE22 3DT, UK; lindsay.snell@nhs.net; 4University Hospitals of Derby and Burton NHS Foundation Trust, Obstetrics and Gynaecology, Derby DE22 3DT, UK; shaimaa.aboeldalyl@nhs.net

**Keywords:** obesity, PCOS, NLRP3, inflammasomes, caspase-1, ASC, IL-1β

## Abstract

Inflammasomes have recently been implicated in the pathogenesis of several chronic inflammatory disorders, such as diabetes and obesity. The aim of this meta-analysis was to investigate the possible role of the NLRP3 inflammasome in obesity and polycystic ovarian syndrome (PCOS). A comprehensive search of electronic databases was conducted to identify studies investigating NLRP3 its related components (Caspase 1, ASC and IL-1β) in adipose tissue and/or blood from obese individuals compared to non-obese controls. Another search was conducted for studies investigating NLRP3 in PCOS women and animal models. The ssearched databases included Medline, EMBASE, Cochrane Library, PubMed, Clinicaltrials.gov, the EU Clinical Trials Register and the WHO International Clinical Trials Register. The quality and risk of bias for the included articles were assessed using the modified Newcastle–Ottawa scale. Data were extracted and pooled using RevMan software for the calculation of the standardized mean difference (SMD) and 95% confidence interval (CI). Twelve eligible studies were included in the obesity systematic review and nine in the PCOS review. Of the obesity studies, nine (*n* = 270) were included in the meta-analysis, which showed a significantly higher adipose tissue NLRP3 gene expression in obese (*n* = 186) versus non-obese (*n* = 84) participants (SMD 1.07; 95% CI, 0.27, 1.87). Pooled analysis of adipose tissue IL-1β data from four studies showed significantly higher IL-1β gene expression levels in adipose tissue from 88 obese participants versus 39 non-obese controls (SMD 0.56; 95% CI, 0.13, 0.99). Meta-analysis of adipose tissue ASC data from four studies showed a significantly higher level in obese (*n* = 109) versus non-obese (*n* = 42) individuals (SMD 0.91, 95% CI, 0.30, 1.52). Of the nine PCOS articles, three were human (*n* = 185) and six were animal studies utilizing PCOS rat/mouse models. All studies apart from one article consistently showed upregulated NLRP3 and its components in PCOS women and animal models. In conclusion, obesity and PCOS seem to be associated with upregulated expression of NLRP3 inflammasome components. Further research is required to validate these findings and to elucidate the role of NLRP3 in obesity and PCOS.

## 1. Introduction

Obesity and its related health problems have been rapidly growing worldwide since the late 20th century. According to the World Health Organization (WHO), an estimated 650 million adults were affected by obesity worldwide in 2016, a figure predicted to double by 2030 [1]. WHO defines obesity (body mass index (BMI) > 30 kg/m^2^) as excessive adipose tissue accumulation that could harm an individual’s health [1].

Adipose tissue, also known as ‘fat’, is a loose connective tissue that is composed of three main compartments, including adipocytes, non-fat cells (known as stromal vascular fraction—SVF), and extracellular matrix (ECM) surrounding all cells [2,3]. Adipocytes are the main cellular component forming >80% of the tissue volume and are responsible for both energy storage and endocrine activity. The SVF contains heterogeneous cell populations, including inflammatory cells (macrophages), immune cells, preadipocytes, fibroblasts and adipose-derived mesenchymal stem cells (ADSCs), vascular cells, and immune cells. The ECM is made of collagen fibres, blood vessels, and neurons [2,3]. 

The concept of adipose tissue being an inert energy storage tissue has long been challenged, and it is now well recognised as an active organ with numerous metabolic, endocrine, and immunological functions. The excess adipose tissue accumulation in obesity is associated with the release of adipokines, which are hormone-like mediators that control insulin sensitivity and energy balance. Obesity is therefore considered a chronic metabolic condition linked with multiple disorders, including insulin resistance (IR), type 2 diabetes (T2DM), atherosclerosis, cardiovascular complications, and cancer [4]. In women, obesity has also been associated with reproductive dysfunctions, including anovulation and polycystic ovarian syndrome (PCOS) [5]. 

Obesity is considered a chronic inflammatory condition as evidenced by the increased immune cell infiltration of adipose tissue with excess production of circulating proinflammatory factors, including cytokines and C-reactive protein (CRP) [6,7,8,9,10,11]. This systemic chronic inflammation has been implicated in the pathogenesis of many metabolic disorders, such as IR, T2D, and PCOS, in obese subjects [12,13,14,15]. Although numerous studies have reported that obesity-related excess adipose tissue plays a key role in causing the chronic inflammatory state, the exact underlying molecular mechanisms remain unclear [16]. 

Inflammasomes, a term first introduced by Jürg Tschopp in 2002 [17], have recently been implicated in the pathogenesis of several chronic inflammatory and metabolic disorders [18]. They are defined as intracellular multiprotein complexes, which act as immune system receptors and sensors that trigger the activation of potent inflammatory mediators. They regulate the immunological response to exogenous and endogenous stimuli, including pathogen- and danger-associated molecular patterns (PAMPs and DAMPs) through the production of the interleukin-1 family of cytokines [19]. All inflammasomes share the same basic structural template, which consists of three components including an integral pattern recognition receptor (PRR), procaspase-1, and the apoptosis-associated speck-like protein containing a CARD (PYCARD, also called ASC). The PRR family includes nucleotide-binding oligomerization domain (NOD)-like receptors (NLRs), toll-like receptors (TLRs), and retinoic-acid-inducible gene I-like helicases (RLHs) [20,21,22]. 

The NLRP3 inflammasome is the most studied member of the NLR family. Once activated by DAMPs or PAMPs, NLRP3 associates with ASC and recruits procaspase-1. This assembly of the inflammasome complex results in the autoproteolysis of procaspase-1, producing the active caspase-1 subunits that subsequently mediate inflammatory responses with subsequent pyroptosis. Caspase-1 induces the inflammatory process via the cleavage and release of the proinflammatory cytokines IL-1β and IL-18 [23,24,25]. 

Strong evidence indicates that the NLRP3 inflammasome plays a crucial role in obesity-induced inflammation and IR [26,27]. The expression of NLRP3 in subcutaneous and visceral adipose tissue (SAT and VAT) has been positively correlated with BMI and IR in obese subjects [28,29]. Furthermore, NLRP3 inflammasome suppression has been found to reduce obesity-related inflammation and enhance insulin sensitivity [30]. Obesity-related dyslipidemia and lipotoxicity have been shown to serve as sources of endogenous DAMPs that activate the inflammasome [30]. NLRP3 may also be activated in macrophages by several obesity-related factors, including elevated levels of ceramides, reactive oxygen species (ROS), ATP, and mitochondrial dysfunction [31,32,33,34].

In a recent systematic review and meta-analysis, our group provided strong evidence for an association between PCOS and chronic inflammation [15]. Given the well-known mechanistic links between obesity, IR, and PCOS, it is plausible to hypothesize that the NLRP3 inflammasome may play a similar role in inducing chronic inflammation in the AT of PCOS women. 

The role of NLRP3 in obesity-related inflammation has previously been investigated in several studies with conflicting results. While some studies reported increased NLRP3 levels in obese versus lean subjects [35,36], others showed no differences between the two groups [28,37]. Although the role of NLRP3 in obesity has previously been investigated in a systematic review published in 2017 by Rheinheimer et al., the majority of the included papers were animal studies. Furthermore, many of the human studies in this review mainly focused on metabolically unhealthy obese (MUO) participants, making it difficult to draw any firm conclusion [38]. Moreover, since the previous systematic review, many recent human studies assessing the role of NLRP3 in obese individuals have been published. 

To date, the status of the NLRP3 inflammasome and its dynamics in the adipose tissue of PCOS women have not been previously investigated. There are only very few recent human and animal studies investigating the status of NLRP3 in the blood and ovaries of PCOS women and PCOS rodent models. The purpose of the current study was therefore to conduct an updated and a more comprehensive systematic review including the most recent human studies assessing the NLRP3 status in metabolically healthy obese (MHO) subjects. In addition, this study will also review the limited studies investigating the role of NLRP3 in PCOS.

## 2. Materials and Methods

This systematic review has been conducted in line with the Preferred Reporting Items for Systematic Reviews and Meta-Analyses (PRISMA) guidelines and was prospectively registered with PROSPERO (registration number CRD42022334993).

### 2.1. Eligibility Criteria for Study Selection 

#### 2.1.1. NLRP3 and Obesity

We considered all studies comparing the expression levels of the NLRP3 inflammasome and its related components in adipose tissue and blood in obese versus non-obese controls. Only studies using the WHO obesity definition [39] and matching/adjusting for age were eligible for the review. In addition, the review included studies conducted on humans, written in the English language, and involving metabolically healthy participants aged 20–65 years who were drug naïve and with no history of any disease. If more than one article by the same research group was identified, only the study with the larger sample size was included in the meta-analysis to avoid case duplication.

#### 2.1.2. NLRP3 and PCOS

We considered all studies comparing the expression levels of NLRP3 inflammasome and its related components in adipose tissue, blood, or any other tissue between PCOS versus non-PCOS healthy controls. We also searched for animal studies investigating NLRP3 inflammasome in PCOS mouse/rat models. Only studies using the Rotterdam diagnostic criteria for PCOS [40] and matching/adjusting for age were eligible for the review. In addition, the review included studies written in the English language, and involving metabolically healthy women aged 20–45 years who were drug naïve and with no history of any disease. If more than one article by the same research group was identified, only the study with the larger sample size was included in the meta-analysis to avoid case duplication.

### 2.2. Outcome Measures

#### 2.2.1. Main Outcome

NLRP3 expression levels in adipose tissue and blood.

#### 2.2.2. Secondary Outcome Measures

Casp-1, ASC, and IL-1β expression levels in adipose tissue and blood.

### 2.3. Search Strategy

All searches were performed by a clinical librarian (LS) and repeated by a second independent librarian. A further manual search of the references of the selected articles was conducted to identify eligible studies.

#### 2.3.1. NRP3 and Obesity

Medline (Ovid); EMBASE (Ovid); CENTRAL (www.thecochranelibrary.com) accessed on 20 October 2022; PubMed; Clinicaltrials.gov; the EU Clinical Trials Register; and the World Health Organisation International Clinical Trials Register were systematically searched starting from 1946 to October 2022 for pertinent studies. A combination of the following search terms was used: (“overweight”[MeSH Terms] or obesity or overweight or over-weight) AND ((“inflammasomes”[MeSH Terms]) OR inflammasomes OR (“nlr family, pyrin domain containing 3 protein”[MeSH Terms]) OR “NLR family pyrin domain containing 3” OR nlrp3 OR (“caspase 1”[MeSH Terms]) OR “Interleukin-1 beta converting enzyme” OR caspase-1 OR “Apoptosis-associated speck-like protein containing a CARD” OR PYCARD).

#### 2.3.2. NRP3 and PCOS

The above databases were also systematically searched starting from 1946 to April 2023 for pertinent studies related to PCOS and NLRP3. A combination of the following search terms was used: (Polycystic Ovary Syndrome [MeSH Terms] OR polycystic ovar* OR sclerocystic ovar* OR micropolycystic ovar* OR stein leventhal* OR stein-leventhal* OR PCOS) AND (NLR Family, Pyrin Domain-Containing 3 Protein [MeSH Terms] OR “NLR family pyrin domain containing 3” OR nlrp3).

### 2.4. Screening and Selection of Retrieved Studies 

The titles and abstracts of the retrieved articles were screened for relevance to the systematic review. The full texts of the relevant articles were then reviewed for eligibility based on the above inclusion criteria. Two independent reviewers (SAA and SA) conducted this process, and in cases of disagreement, a consensus was reached after a discussion between them.

### 2.5. Assessment of Quality and Risk of Bias

The risk of bias and the quality of included articles were assessed using a modified Newcastle-Ottawa scale [41,42]. Three main aspects were evaluated in each study namely selection, comparability, and outcome with a maximum scores of four, two and three, respectively. The selection was assessed based on the accurate definition of cases, the inclusion of consecutive cases, power calculation and the inclusion of matched healthy controls. Studies received up to two stars in comparability if the main confounder namely the age of participants was properly matched or adjusted for. Three aspects of outcome were evaluated including a clear description of the methods used for the measurements of NLRP3 and related components, validated laboratory methods, and clear statistical comparisons between obese and non-obese groups.

### 2.6. Data Extraction and Analysis

Data (mean ± SD) were extracted from the individual articles including sex, age, BMI, weight, NLRP3, IL-1β, and ASC. Where missing data were encountered, the authors were contacted to provide these data. The authors of two articles responded with the missing data [43,44]. 

In eight studies, NLRP3 data were presented as graphs with no exact numbers given. Of these, six studies presented data graphically as mean ± SEM [36,37,45,46,47,48,49,50], one as mean and *p*-value [28] and another as median (IQR) [35]. We first extracted the data from the graphs using WebPlotDigitizer (version 3.10, http://arohatgi.info/WebPlotDigitizer/ (accessed on 12 June 2023) [51]. This programme was recommended by [52] and recently validated by [53] for use in meta-analysis to capture study data from graphs. We then calculated the mean ± SD from the mean ± SEM, mean and *p*-value or median [IQR] according to methods described by Higgs and Deeks in the Cochrane Handbook for Systematic Reviews [42].

The extracted data were uploaded into RevMan software, version 5.9 (The Nordic Cochrane Centre, Copenhagen, Denmark; The Cochrane Collaboration, 2011) for meta-analysis. The data were pooled and the standardized mean difference (SMD) and 95% confidence interval (CI) were calculated for NLRP3, IL-1β, and ASC. We used the SDM model in this meta-analysis in view of the variations in the NLRP3 measurements used across the included studies [54]. The SMD is the MD divided by the standard deviation (SD), derived from either or both of the groups [55]. In addition to being independent of the unit of measurement, the SMD has been found to be more generalizable and an easier way to judge the magnitude of difference between groups [56]. The general rule described by Cohen et al. suggests that an SMD of 0.2 represents a “small” difference, an SMD of 0.5 represents a “medium” difference, and an SMD of 0.8 represents a “large” difference [55]. 

I-squared (I2) statistics and the chi-square test were used to evaluate the statistical heterogeneity between studies. High heterogeneity was indicated by I2 ≥ 50% or chi-square analyses higher than its degree of freedom [57]. For the meta-analysis, a random effects model was applied when heterogeneity was considerable.

## 3. Results

### 3.1. Search Results—NLRP3 and Obesity

A total of 3173 articles were retrieved following the initial search of the electronic databases, reduced to down to 1810 after the exclusion of duplicates. Of these, 1773 irrelevant articles were excluded during the screening of titles and abstracts. After full text review of the remaining 37 papers, a further 25 articles did not meet the eligibility criteria and were therefore excluded. The Reasons for exclusions are summarised in Figure 1. The remaining 12 studies [28,35,36,37,43,44,45,46,47,48,49,50] fulfilled the eligibility criteria and were included in this review (Figure 1).

#### 3.1.1. Risk of Bias and Quality Assessment of Selected Studies

Table 1 shows the quality scores of the 12 studies included in the NLRP3 and obesity review. Nine studies [28,35,36,37,40,42,43,44,45] scored ≥ 7, while the remaining three articles [39,41,46] scored 6 on the modified Newcastle–Ottawa scale.

#### 3.1.2. Excluded Studies

Of the twenty-five excluded articles, six did not include the primary outcome, ten did not measure inflammasomes in adipose tissue or blood, two were animal studies, and seven included metabolically unhealthy participants.

#### 3.1.3. Included Studies

The review included 12 human studies (*n* = 334) investigating NLRP3 in obese (*n* = 218) versus non-obese (*n* = 116) participants. Of these, 11 studies also provided data on other NLRP3 components including IL-1β, ASC, CASP1, and IL-18. The characteristics of all the studies are summarised in Table 2.

#### 3.1.4. Study Designs

Of the 12 included studies in this review, eleven were observational case–control and one was an interventional study that measured the effect of bariatric surgery on the inflammasome levels (Table 2). The preoperative baseline data of the interventional study were suitable for this review [45].

#### 3.1.5. Study Participants

An appropriate selection of participants was used in all studies, which met our inclusion criteria. All participants were in the age range between 33 and 63 years, had no metabolic or endocrinological diseases, and were not on any medication that might affect inflammatory status. In most studies, the ages of the obese and non-obese participants were comparable, apart from two studies where there was a difference > 10 years in mean/median age between the two groups [44,45]; both studies scored low on the comparability assessment. Participants were weighed after an overnight fast and samples were taken in the morning to ensure comparability.

### 3.2. Search Results—NLRP3 and PCOS

Our search retrieved 67 articles, of which 56 were deemed irrelevant after screening the titles and abstracts. The remaining 11 studies were reviewed in full.

#### 3.2.1. Excluded Studies 

Of the 11 identified articles, two studies were excluded, including a human study which did not measure NLRP3 expression levels, but investigated the frequencies of NLRP3 single nucleotide gene polymorphisms in leukocytes of PCOS and control women [58]. The other article was an animal study, which measured NLRP3 as part of a complex in combination with Thioredoxin-interacting protein in ovarian tissue of the PCOS model with no separate data for NLRP3 [59].

**Table 2 ijms-24-10976-t002:** Characteristics of the included studies investigating NLRP3 in obese individuals.

1st Author, Year	Country	Number of Participants		BMI		Age		Adipose Tissue/PBMCs	NLRP3	NLRP3 Related Components
		Non-Obese	Obese	Non-Obese	Obese	Non-Obese	Obese			
Esser, 2013 [28]	Belgium	9	21	22.7 ± 0.9	42.3 ± 1.0	42.0 ± 4.0	35.0 ± 2.0	SAT/VAT	ND	IL-1β, Caspase 1, ASC
Goossens, 2012 [35]	France	9	10	23.4 ± 0.4	34.2 ± 1.3	59.2 ± 2.55	59.6 ± 3.1	SAT	↑	IL-1β/IL-18/Caspase 1
Yin, 2014 [36]	US	12	24	22.0 ± 0.7	36.8 ± 1.2	54.3 ± 2.4	59.1 ± 1.8	SAT	↑	IL-1β/ Caspase 1/ASC
Serena, 2016 [37]	Spain	4	4	21.9 ± 1.9	33.1 ± 2.1	45.0 ± 7.6	39.0 ± 8.9	SAT/VAT	ND	IL-1β/Caspase 1
Dalmas, 2014 [45]	France	4	16	26.1 ± 1.4	47.5 ± 1.2	60.1 ± 4.6	41.6 ± 1.9	SAT/VAT	↑	Pro-IL-1β
Pandolfi, 2015 [46]	Argentina	8	6	22.0 ± 0.6	34.0 ± 2.6	40.0 ± 2.9	41.0 ± 3.5	VAT	ND	IL-1β
Oliveira, 2020 [43]	Brazil	8	24	21.4 ± 1.9	33.6 ± 3.3	43.0 (29.0–59.2)	46.5 (33.0–58.0)	SAT	ND	IL-1β/IL-18
Beals, 2021 [48]	US	13	21	23.1 ± 0.5	37.8 ± 1.1	38.0 ± 2.0	40.0 ± 1.0	SAT	↑	-
Frühbeck, 2021 [49]	Spain	11	34	20.8 ± 0.5	38.2 ± 1.1	45.0 ± 6.0	49.0 ± 2.0	VAT	↑	ASC, IL-1β/IL-18/
Unamuno, 2021 [50]	Spain	10	30	22.1 ± 3.0	42.5 ± 5.2	36.0 ± 3.0	40.0 ± 2.0	VAT	↑	IL-1β/IL-18/ASC
Liu, 2018 [47]	China	20	20	22.0 ± 0.5	28.4 ± 0.4	42.3 ± 5.3	45.9 ± 4.9	**PBMCs**	↑	CASP1
Williams, 2022 [44]	Australia	8	8	22.6 (21.7–24.1)	41.4 (37.4–49.3)	27.8 (24.1–35.2)	39.4 (35.1–45.4)	**PBMCs**	↑	IL-1β/Caspase 1

Abbreviations: PBMCs, Peripheral blood monocytes; SAT, Subcutaneous adipose tissue; VAT, Visceral adipose tissue, ND, No difference.

#### 3.2.2. Included Studies 

Given the scarcity of human studies, we decided to include both human and animal articles. Nine studies were deemed eligible and were included in the systematic review including three human studies [60,61,62] and six animal studies [63,64,65,66,67,68]. Two of the human studies measured NLRP3 in PBMC and one in luteinised granulosa cells (Table 3). The animal studies investigated NLRP3 and its related components in ovarian tissue, blood, and/or granulosa cells obtained from PCOS mouse/rat models (Table 3).

### 3.3. Systematic Review—NLRP3 and Obesity 

#### 3.3.1. NLRP3 Inflammasome

NLRP3 gene expression levels were measured in all 12 studies (*n* = 334) that compared levels in obese versus non-obese controls, of which 10 measured NLRP3 in SAT/VAT and two in PBMCs (Table 2 and Table 4). Six adipose tissue studies (*n* = 194) reported significantly increased levels of NLRP3 in obese (*n* = 135) versus control (*n* = 59) participants [35,36,45,48,49,50]. The remaining four studies showed no significant difference between NLRP3 levels in the adipose tissue of obese versus non-obese participants [28,37,43,46]. However, all four studies reported an increase in adipose tissue NLRP3 expression levels in metabolically unhealthy obese participants (MUO) compared to metabolically healthy obese (MHO) and lean participants. 

Of the 10 adipose tissue studies, four (*n* = 121) measured NLRP3 gene expression in SAT only [35,36,43,48], three (*n* = 99) in VAT only [46,49,50], and three (*n* = 58) in both SAT and VAT [28,37,45]. Of the seven studies (*n* = 179) measuring SAT NLRP3, four (*n* = 89) showed significantly increased levels of NLRP3 in obese compared to non-obese participants [35,36,45,48] and three (*n* = 70) reported no significant difference between the groups [28,37,43].

Of the six studies (*n* = 157) measuring NLRP3 gene expression in VAT, three (*n* = 105) showed significantly increased NLRP3 levels in obese versus non-obese controls [45,49,50], and three (*n* = 52) showed no significant difference between the groups [28,37,46]. 

Two of the studies [28,29], which found that NLRP3 was upregulated in both VAT and SAT in obese individuals, also reported that NLRP3 expression was positively correlated with BMI. Two studies (*n* = 56) reported NLRP3 gene expression levels in PBMCs showing significantly higher levels in obese (*n* = 40) versus non-obese (*n* = 16) participants [44,47]. 

#### 3.3.2. IL-1ß

IL-1ß gene expression levels were measured in 10 studies including nine in adipose tissue and one in PBMCs. Seven of the nine adipose tissue studies in addition to the PBMCs study reported a statically significant increase in IL-1ß expression levels in obese participants compared to controls [35,36,37,39,44,45,46,49,50], while the remaining two adipose tissue studies showed no difference in levels between the two groups [28,43]. 

#### 3.3.3. Caspase 1

Caspase 1 gene expression levels were measured in six articles (four in adipose tissue and two in PBMCs), of which five reported significantly increased levels in obese versus non-obese participants [35,36,37,44,47], while one study [28] showed no statistically significant difference between the two groups.

#### 3.3.4. ASC

ASC gene expression levels were measured in adipose tissue in four studies, of which three reported significantly higher levels in obese versus non-obese participants [36,49,50], while the remaining study found no statistically significant difference between the two groups [28].

**Table 3 ijms-24-10976-t003:** Characteristics of the included studies investigating NLRP3 in PCOS women.

Human Studies											
First Author, Year	Country	Study Design	Number of Participants		BMI		Age		Blood/ Tissue	NLRP3	NLRP3 Related Components
			PCOS	Control	PCOS	Control	PCOS	Control			
Rostamtabar, 2020 [62]	Iran	Case–control	30	30	*	**	(20–40)	(20–40)	PBMC	↑	ASC, IL-1β, IL-18,
Guo, 2020 [60]	China	Case–control	38	30	ND	ND	ND	ND	PBMC	↑	Casp-1, IL-1β, IL-6, TNF-α
Liu, 2021 [61]	China	Case–control	27	30	24.5 ± 2.8	19.3 ± 0.9	27.2 ± 3.3	29.5 ± 3.4	LGCs	↑	Casp-1, ASC, IL-1β, IL-18/
**Animal Studies**											
**First Author, Year**	**Country**	**PCOS Model**	**Number of Animals**			**Treatment**	**Other Tests**	**mRNA,** **Protein, IH**	**Blood/** **Tissue**	**NLRP3**	**NLRP3 Related Components**
			**PCOS**	**Control**	**Treated PCOS**						
Wang, 2017 [68]	China	Letrozole-induced rat	ND	ND	ND	DMBG	CRP, LPO, SOD, Catalase	mRNA	Ovary	↑	IL-1β, ASC, Casp-1,
Ibrahim, 2022 [64]	Egypt	Letrozole-induced rat	12	12	12	Diacerein/Metformin	TNF-α, NFκB	mRNA	Ovary	↑	IL-1β
Li, 2021 [65]	China	DHEA-induced mouse	ND	ND	ND	MiR-1224-5p	TNF-α	Protein	Ovary	↑	Casp-1, ASC, IL-1β
Wang, 2020 [67]	China	Testosterone-induced mouse	16	16	7	INF39	IL-6, TLR4, NLRP1, NLRC3	mRNA/ Protein	GCs	↑	Casp-1, IL-18, IL-1β
Cai, 2022 [63]	China	DHEA-induced mouse	ND	ND	ND	Plumbagin	GSDMD, NEK7	mRNA/Protein	GCs	No changed	Casp-1, ASC, IL-18, IL-1β
Olaniyi, 2022 [66]	Nigeria	Letrozole-induced rat	6	6	6	Acetate	GT/leptin, GSDMD adiponectin, NF-kB, TNF-α, Kisspeptin	IH	VAT, HPT	↑	Nil

The data are presented as mean ± sd; (range). * BMI < 25, *n* = 8, and ≥25 *n* = 22); ** BMI < 25, *n* = 19, and ≥25 *n* = 11. Abbreviations: ND, no data; PBMC, peripheral blood mononuclear cell; LGCs, luteinised granulosa cells; GCs, granulosa cells; VAT, visceral adipose tissue; HPT, hypothalamus; GT, glucose tolerance; GSDMD, gasdermin; DLOP, lipid peroxidation; SOD, activity of superoxide dismutase; DHEA, dehydroepiandrosterone; DMBG, dimethyldiguanide; IH, immunohistochemistry.

**Table 4 ijms-24-10976-t004:** NLRP3, Casp-1, ASC, and IL-β expression levels in AT and PBMCs from obese individuals in all 12 studies included in the systematic review.

First Author, Year	Number of Participants		NLRP3			Caspase-1			ASC			IL-1β		
	Non-Obese	Obese	Non-Obese	Obese	*p*	Non-Obese	Obese	*p*	Non-Obese	Obese	*p*	Non-Obese	Obese	*p*
**SAT/VAT**														
Beals, 2021 [48]	13	21	2.07 ± 0.53	3.10± 0.50	<0.05	-	-	-	-	-	-	-	-	-
Dalmas, 2014 [45]	4	16	1.04 ± 0.42	2.83 ± 2.50	<0.05	-	-	-	-	-	-	1.10 ± 0.72	3.06± 2.40	-
Esser, 2013 [28]	9	21	0.98 ± 1.20	1.12 ±1.37	<0.05	-	-	-	1.01 ± 1.22	1.31 ± 1.60	<0.05			-
Frühbeck, 2021 [49]	11	34	2.30 ± 1.89	12.11 ± 2.21	<0.01	-	-	-	1.80 ± 1.52	2.66 ± 1.34	<0.01	3.08 ± 0.99	2.500 ± 1.74	-
Goossens, 2012 [35]	9	10	0.56 ± 0.37	0.82 ± 0.52		6.18 ± 2.65	12.26 ± 6.70	<0.01	-	-	-	0.38 ± 0.51	0.50 ± 0.67	-
Oliveira, 2020 [43]	8	24	1.46 ± 1.28	1.49 ± 1.19	NS	-	-	-	-	-	-	1.562 ± 1.19	1.765 ± 1.860	-
Pandolfi, 2015 [46]	8	6	0.63 ± 0.39	0.45 ± 0.48	<0.01	-	-	-	-	-	-	0.65 ± 0.90	18.48 ± 9.79	-
Serena, 2016 [37] *	4	4	-	-		-	-	-	-	-	-	-	-	<0.001
Unamuno, 2021 [50]	10	30	0.99 ± 0.63	3.73 ± 2.19	<0.01	-	-	-	1.00 ± 0.44	1.742 ± 0.62	<0.01	1.02 ± 1.45	2.16 ± 1.53	-
Yin, 2014 [36] **	12	24	1.06 ± 1.66	6.97 ± 6.85	<0.05				0.99 ± 0.55	5.11 ± 2.98	<0.05	1.00 ± 1.80	2.836 ± 1.71	<0.05
**PBMCs**														
Liu, 2018 [47]	20	20	0.98 ± 1.14	2.68 ± 3.12	<0.01	0.92 ± 1.07	3.33 ± 3.88	<0.01						
Williams, 2022 [44]	8	8	0.004 ± 0.003	0.13 ± 0.102	<0.05	0.005 ± 0.005	0.97 ± 1.13	NS						

The data are presented as mean ± SD. Eleven studies reported NLRP3 gene expression levels and one study [36] reported protein expression. * The data are presented graphically in this study and could not be retrieved from the figures. ** The NLRP3 data are protein expression measured by Western blotting, with no gene expression data. Abbreviations: **SAT**, subcutaneous adipose tissue; **VAT**, visceral adipose tissue; **PBMCs**, peripheral blood monocytes; **NS**, nonsignificant.

#### 3.3.5. IL-18

Adipose tissue IL-18 gene expression levels were measured in four studies, of which three reported significantly higher levels of IL-18 in obese versus non-obese individuals [35,49,50], while in one study there was no significant difference in IL-18 levels between the two groups [43].

### 3.4. Systematic Review—NLRP3 and PCOS

#### 3.4.1. Human Studies

Three studies investigated NLRP3 and its related components in PCOS women [60,61,62] (Table 3). Of those, two measured NLRP3 expression levels in PBMC [60,62] and one measured it in stimulated luteinised granulosa cells (LGCs) obtained during *in-vitro fertilisation* (IVF) [61]. 

Both Rostamtabar et al. and Guo et al. reported remarkably higher levels of NLRP3 inflammasomes in the PBMCs of PCOS (*n* = 68) versus control women (*n* = 60) [60,62]. Rostamtabar et al. showed significantly higher mRNA expression levels of caspase-1, ASC, and absent in melanoma 2 (AIM2) in PBMCs from PCOS women (*n* = 30) versus controls (*n* = 30) [51]. Additionally, they reported higher serum levels of IL-18, but not IL-1β in PCOS women versus controls. They showed a positive correlation between NLRP3 and AIM2 expression levels with IL-18 (*r* = 0.59, *p* < 0.0001; and *r* = 0.83, *p* < 0.032), but not with IL-1β. While IL-1β serum levels correlated positively with BMI in PCOS women, there was no such correlation in non-PCOS participants. They found no changes in other inflammasomes including NLRP1, NLRP12, NAIP, and NLRC4. Similarly, Guo et al. reported higher levels of mRNA expression levels of NRP3 and caspase-1 and protein levels of NLRP3 in the PBMCs of PCOS (*n* = 38) versus control (*n* = 30) women [49]. They also found higher NLRP3 and caspase-1 in PCOS women with comorbid psychological disorders (PD) versus PCOS without PD (*n* = 30). Additionally, they evaluated the effect of 12-week treatment with insulin-sensitising drugs on NLRP3 expression levels in PBMC. The results showed that 12-week treatment with pioglitazone metformin complex (PM), but not with metformin, resulted in a reduction in NLRP3, caspase-1, IL-1β, IL-6, and TNF-α. The results also showed that PM significantly reduced the SCL-90-R scores of anxiety and depression in PCOS women with comorbid psychological distress. The authors concluded that PM alleviates psychological distress by inhibiting NLRP3. However, the fact that PM reduces psychological distress and NLRP3 activity does not necessarily indicate a mechanistic relationship between these two effects.

Liu et al. measured gene and protein expression levels of NLRP3 and its related components in luteinised granulosa cells (LGCs) obtained from 27 PCOS and 30 non-PCOS women undergoing IVF [61]. They also measured concentration levels of IL-1β and IL-18 in follicular fluid obtained from the two groups during IVF. They reported significantly higher gene and protein expression levels of all parameters in the LGCs including NLRP3, caspace-1, ASC, and IL-1β. Follicular levels of IL-1β and IL-18 were also higher in PCOS versus control women. In addition, they showed elevated levels of LGCs TLR4 and IL-6 in LGCs from PCOS women compared to controls. They concluded that the altered inflammatory microenvironment in the follicular fluid of PCOS women might provide a novel mechanism for the inflammatory process of PCOS [61]. However, it is difficult to draw any conclusion from this study as NLRP3 was measured in stimulated GCs, which may not reflect the native status in the unstimulated ovary. 

It is worth noting that neither of the above three studies matched their PCOS and healthy participants or adjusted their results for BMI (Table 3) [60,61,62]. Both Rostamtabar et al and Liu et al included more obese subjects in the PCOS groups compared to the controls (Table 3) [61,62]; while Gue et al. did not report the BMI data for either of the study groups [60].

#### 3.4.2. Animal Studies

Six studies investigated NLRP3 and its related components in PCOS mouse/rat models (Table 3). Five of these studies reported enhanced expression levels of NLRP3 and its related components in the PCOS animal model, but one study showed no change in NLRP3 [63]. Three studies measured the inflammasome in ovarian tissue [64,65,68], two in granulosa cells (GCs) [63,67] and one in the hypothalamus and VAT [66] of the rat/mouse models. Three studies used PCOS mouse [65,66,68] and three used PCOS rat models [63,64,67]. Three studies induced the PCOS model by treating the animals with letrozole [64,66,68], two with DHEA and one with testosterone (T). They all reported the success of PCOS development in the treated animals.

##### NLRP3 in Ovarian Tissue

All three studies using ovarian tissues reported increased NLRP3 and its related components in PCOS mouse/rat models compared to control animals [64,65,68].

Wang et al., reported increased NLRP3 and ASC mRNA levels in the ovarian tissue of letrozole-induced PCOS rats, which were significantly decreased after metformin treatment [68]. While Procaspase 1 expression was not changed, active caspase 1 expression was dramatically increased in PCOS ovaries, which was decreased significantly after metformin treatment. Additionally, the authors reported a significant increase in caspase 1 activity and IL-1β production in PCOS ovaries, with both reduced after metformin treatment. They also showed enhanced oxidative stress in PCOS ovaries, which was reduced by metformin. They concluded that PCOS seems to be associated with enhanced oxidative stress and activation of NLRP3 inflammasomes, which may lead to chronic low-grade inflammation. Metformin seems to have a protective effect with the reduction in oxidative stress and NLRP3 inflammasomes in PCOS women.

Ibrahim et al, reported significantly higher (6-fold) mRNA levels of NLRP3 and caspase 1 in ovarian tissue of letrozole-induced PCOS rats than in normal controls [64]. They also found higher ovarian and serum levels of IL-1β and higher serum levels of TNF-α and IL-6 in the PCOS rats compared to control rats. They investigated the effect of diacerein (DIA) (in different doses) and metformin on inflammasomes in PCOS and control rats. All the DIA and metformin treated groups showed significantly reduced NLRP3, caspase 1, IL-1β, TNF-α and IL-6, with greater reduction when using DIA than metformin. Only the treatment with DIA-25 and DIA-50 groups showed normalized values. The authors concluded that NLRP3 inflammasomes are enhanced in PCOS rats, an effect that can be ameliorated by DIA and to a lesser extent by metformin. 

Li et al, reported higher protein expression levels of NLRP3, caspase-1, ASC, IL-1β and TNF-α, but unchanged Procaspase 1 levels in ovarian tissue obtained from the DHEA-induced PCOS mouse model versus control mice [65]. They also found that enhanced activation of the NLRP3 inflammasome was accompanied by the downregulation of miR-1224-5p in the ovarian tissue of PCOS mice. They also carried out other experiments on human granulosa-like tumor (KGN) cells, which showed that miR-1224-5p could inhibit NLRP3 inflammasome activation. They concluded that the study suggested that miR-1224-5p might be a promising target for treating PCOS [65]. 

##### NLRP3 in Granulosa Cells

Of the two studies measuring NLRP3 in GCs, one showed an elevation of NLRP3 [67], while the other showed no change in PCOS mice [63]. Wang et al. reported markedly enhanced gene and protein expression levels of TLR4 and NLRP3 in the GCs obtained from T-induced PCOS versus control mice [67]. They also reported increased protein expression levels of ASC, caspase-1, IL-1β, and IL-18 in the PCOS mouse model versus controls. On the other hand, there was no difference in NLRP1 and NLRC4 mRNA levels in GCs between both groups. Additionally, they showed that treatment of GCs with dihydrotestosterone (DHT) induced cell death with an associated marked increase in NLRP3, caspase-1, ASC, IL-1β, and IL-18 in GCs. They concluded that hyperandrogenism could trigger NLRP3 inflammasome activation resulting in low-grade inflammation in PCOS mice. They also showed that activation of the NLRP3 inflammasome accelerates ovarian fibrosis in the PCOS mice model. When GCs were treated with INF39 (a specific NLRP3 inhibitor) it resulted in the suppression of ovarian fibrosis indexes (alpha-smooth muscle actin (α-SMA), connective tissue growth factor (CTGF), and TGF-β) and remarkably reduced ovarian interstitial fibrosis. Cai et al. reported a marked increase in ASC, but no change in NLRP3 mRNA and protein in GCs from DHEA-induced PCOS mice [63]. The elevated ASC levels were reduced by plumbagin treatment. In addition, they reported that the treatment of the PCOS mice with a caspase-1 inhibitor (Vx-765) significantly reduced the development of PCOS changes in the ovary based on H&E staining [63]. They also carried out further experiments showing that plumbagin can reduce the NLRP3-induced pyroptosis of GCs. However, the authors did not explain or discuss the discrepancy between the high levels of ASC with no change in NLRP3. 

##### NLRP3 in VAT and Hypothalamus 

Olaniyi et al, measured NLRP3 by immunohistochemistry in VAT and hypothalamus obtained from three groups of rats including untreated control, Letrozole-induced rat model and acetate-treated PCOS rats [66]. They reported no expression of NLRP3 in the VAT or hypothalamus of the control group and the acetate treated PCOS groups, while there was a moderate NLRP3 expression in the VAT and marked expression in hypothalami of PCOS rats. They concluded that PCOS in rat models is associated with increased NLRP3 in VAT and the hypothalamus, an effect that seems to be repressed by acetate [66].

### 3.5. Meta-Analysis—NLRP3 and Obesity 

#### 3.5.1. Main Outcome: NLRP3 Inflammasome 

##### Overall NLRP3 Pooled Analysis

Pooled analysis of nine studies (*n* = 270) with relevant data showed a statically significant increase in SAT/VAT NLRP3 gene expression in obese (*n* = 186) compared to non-obese (*n* = 84) participants (SMD 1.07, 95% CI, 0.27, 1.87; *z* = 2.63; *p* = 0.009; I^2^ = 97%). The heterogeneity between the studies was high (Figure 2). 

##### Sensitivity Analysis: NLRP3 Gene Expression in SAT/VAT

Pooled analysis of SAT/VAT NLRP3 data from eight studies (*n* = 250) scoring ≥ 7 on the modified Newcastle–Ottawa scale showed significantly higher expression levels in obese (*n* = 170) versus non-obese (*n* = 80) participants (SMD 1.11, 95% CI, 0.23, 1.99; *z* = 2.63; *p* = 0.009; I2 = 97%). The heterogeneity between the studies was high (Figure 3).

##### Subgroup Analysis: NLRP3 in SAT and VAT

Data from five studies (*n* = 149) using VAT samples, showed no statically significant difference in NLTP3 gene expression levels between obese (*n* = 107) and non-obese (*n* = 42) participants (SMD 1.25, 95% CI, −0.22, 2.71; *z* = 1.67; *p* = 0.09; I*^2^* = 91%). The heterogeneity between the studies was high (Figure 4). 

Four studies (*n* = 121) using SAT samples showed significantly higher NLRP3 gene expression levels in obese (*n* = 79) versus non-obese participants (*n* = 42) (SMD 0.89, 95% CI, 0.09, 1.68; *z* = 2.17; *p* = 0.03; I*^2^* = 73%). The heterogeneity between the studies was high (Figure 4).

#### 3.5.2. IL-1β Meta-Analysis

Four studies reported adipose tissue IL-1β data suitable for meta-analysis. Pooled analysis of these studies showed significantly higher IL-1β gene expression levels in adipose tissue from 88 obese participants versus 39 non-obese controls (SMD 0.56, 95% CI, 0.13, 0.99; *z* = 2.53; *p* = 0.01; *I*^2^ = 70%) (Figure 5). 

#### 3.5.3. Meta-Analysis of ASC

Four studies (*n* = 151) presented adipose tissue ASC data suitable for meta-analysis. Pooled analysis of these studies showed a statistically significant increase in ASC gene expression in obese (*n* = 109) versus non-obese (*n* = 42) individuals (SMD 0.91, 95% CI, 0.30, 1.52; *z* = 2.92; *p = 0.004; I*^2^ = 61%). The heterogeneity between the studies was high (Figure 6). 

## 4. Discussion

To the best of our knowledge, this is the first meta-analysis that includes human studies investigating the status of the NLRP3 inflammasome in metabolically healthy obese individuals. A total of 12 articles (*n* = 334) were included in the systematic review, of which, 10 measured NLRP3 in adipose tissue and two in PBMCs. Nine of the 10 adipose tissue studies were eligible for meta-analysis (*n* = 270). Although the systematic review of the NLRP3 studies showed conflicting results, most articles (six adipose tissue and two PBMCS studies) reported raised NLRP3 gene expression levels in obese participants compared to control, while the remaining four studies showed no change in NLRP3 in obese individuals. Overall meta-analysis of the nine adipose tissue studies showed a significantly higher SAT/VAT NLRP3 in obese versus non-obese participants. However, NLRP3 subanalysis according to the type of adipose tissue showed increased levels in SAT only. Systematic review of the other NLRP3-related components showed that most studies reported increased gene expression levels of IL-1ß (eight of ten studies), caspase-1 (five of six studies), ASC (three of four studies) and IL-18 (three of four studies). Pooled analysis of IL-1β and ASC data showed significantly higher adipose tissue gene expression levels in obese versus non-obese participants. 

We also present the first systematic review of studies assessing NLRP3 in PCOS. We included three human studies and six animal studies utilising PCOS-induced rat/mouse models. All studies apart from one article consistently showed an increase in NALRP3 and its components in PCOS women as well as PCOS animal models. 

### 4.1. Comparison with Previous Studies 

Our results on obesity and NLRP3 are in agreement with a previous systematic review published in 2017 by Rheinheimer and co-workers who reported that obesity is associated with increased adipose tissue NLRP3 expression levels [38]. However, their results were based on heterogenous findings from human (8 studies) and animal models (11 studies). Furthermore, in contrast to the previous review, which included studies involving MUO individuals and animal models, our study was limited to human studies including only MHO individuals. In the previous review, the inclusion of MUO individuals with various metabolic abnormalities such as diabetes and insulin resistance made it difficult to interpret the results due to the major confounding effect of these diseases on the inflammatory status of the studied individuals. On the contrary, in our review, we included only metabolically healthy obese participants to ensure a firmer conclusion. Furthermore, Rheinheimer et al., only used the PubMed and Embase search engines to identify studies, whereas we conducted a more comprehensive search using all available electronic databases. Moreover, we performed risk of bias assessment of included studies to identify high quality articles for a sensitivity analysis, which was not performed in the previous systematic review. Additionally, our review included more recent human studies that have been published since the last review. We also performed a meta-analysis and investigated other NLRP3-related components including ASC, caspase 1, IL-1β, amd IL-18, which were not evaluated in the previous systematic review. 

Although, our meta-analysis showed upregulated levels of NLRP3 in obese versus non-obese individuals, four of the 10 adipose tissue studies included in our review showed no difference in NLRP3 expressions levels between the two groups. Additionally, another study reported that bariatric-induced weight loss in obese patients had no impact on the expression of NLRP3 [69]. A possible explanation for this discrepancy in NLRP3 results between studies could be the type of tissue (VAT or SAT) and/or the variation in the obesity range used in these studies. Adipose tissue is known for its heterogenous nature and thus poses a challenge to handling and extracting, RNA which is the basic step in PCR reactions used to assess the expression of different genes. 

### 4.2. Limitations and Strengths 

The main limitation of this review is the small size of many of the reviewed studies and the high heterogeneity between studies, which may limit the validity of the evidence. Furthermore, none of the included studies correlated between BMI and inflammasomes levels due to the smaller sample sizes. Another limitation is the lack of data on NLRP3 protein expression in most of the included studies, which only measured gene expression. Arguably, it may be the levels of proteins and their interactions that reflect the status of NLRP3 and its activity in the tissue. It is crucial to understand posttranscriptional regulatory factors, which might play an essential role in the development of inflammation and its related metabolic syndrome. However, protein expression levels do not always correlate with the gene levels and in some cases, they may be lower than expected due to their overactivity leading to a decrease in their levels.

The main strength of this review is the comprehensive literature search, which was conducted by a qualified clinical librarian and independently repeated by a second librarian. The search involved an extensive list of all available databases followed by a manual search of selected papers.

The results of the very few human studies included in this review on PCOS and NLRP3 with poor quality should be considered only preliminary. Furthermore, as mentioned above, none of the human studies matched their study groups or adjusted their results for BMI. As obesity is a major determinant of NLRP3 expression, the results of these studies should be interpreted with extreme caution. Moreover, there have been no studies investigating NLRP3 in adipose tissue. Finally, the use of PCOS mouse models in the animal studies limits the extent to which the findings can be applied in humans.

### 4.3. Interpretation of the Results 

In our review, we established a possible link between adiposity and expression of the NLRP3 inflammasome and its related components; however, given the detailed limitations of this review, this link needs to be further investigated. 

NLRP3 is the cornerstone of adipose tissue chronic inflammation in obesity and insulin resistance. Molecular activation of this inflammatory state is deemed to be caused by overnutrition in obesity [70]. 

It was surprising to see that the pooled data analysis showed that NLRP3 was only upregulated in SAT, but was unchanged in VAT. This is contradictory to the well-established notion that VAT is deemed more inflammatory and active than SAT. This is evidenced by the high number of pro-inflammatory cells in VAT [71,72,73] and the increased amounts of VAT-associated circulating inflammatory markers [74,75,76]. Thus, it is possible that visceral fat is the source of circulating low-grade inflammation, which might be important for the development of lifestyle-related chronic diseases [77,78,79]. Given the small size of the VAT studies included in our review and the conflicting results between these studies, further research is needed to establish the NLRP3 status in VAT. 

Several studies have established the relationship between raised inflammasome markers and various metabolic diseases including DM, although the causal relationship is yet to be determined. In our review, we provide good evidence of ongoing inflammatory status in obese individuals without metabolic abnormalities when compared to non-obese participants, which might lead to the conclusion that inflammation might be responsible for the development of metabolic disorders in obese individuals. Further studies with larger sample sizes are needed to establish this correlation further.

### 4.4. Implications of the Findings 

#### 4.4.1. Research Implications 

Human studies are required to further assess the role of the NLRP3 inflammasome and its related components in inducing chronic inflammation in the adipose tissue of obese and PCOS individuals. Future studies should have larger samples and measure protein as well as the gene expression of all inflammasome NLRP3 components in both SAT and VAT. These studies should also investigate possible molecular mechanisms that trigger inflammasome activities in obese individuals. In addition, future research could also explore possible NLRP3 inhibitory molecules and their therapeutic potential in obese individuals. Future studies should focus on the development of NLRP3 structure-guided direct inhibitors with improved specificity and potency.

#### 4.4.2. Clinical Implications 

It is too early for any clinical application based on our current knowledge of NLRP3 inflammasomes. However, it is envisaged that NLRP3 inhibitors could be developed in the near future for use as effective therapeutics for obesity-associated metabolic and cardiovascular disorders. At present, the only available anti-inflammasome drugs are those that block IL-1β [78]. However, IL-1β is not the only cytokine activated by NLRP3 inflammasome; but other cytokines, such as IL-18, are also activated, which may contribute to NLRP3-associated disorders [80,81]. Furthermore, inhibitors aimed at IL-1β can result in unintentional immunosuppressive effects [82]. Therefore, pharmacological inhibitors, which specifically target the NLRP3 inflammasome only, could be a better option for the treatment of NLRP3-associated conditions.

## 5. Conclusions

Our meta-analysis provides further evidence for the upregulation of the NLRP3 inflammasome and its related components in obese and PCOS individuals, which is indicative of a possible role of this inflammasome in obesity and PCOS-related chronic inflammation. However, given the limited number, small sample sizes, and heterogeneity of the reviewed studies, further research is required to elucidate the role of NLRP3 in obesity.

## Figures and Tables

**Figure 1 ijms-24-10976-f001:**
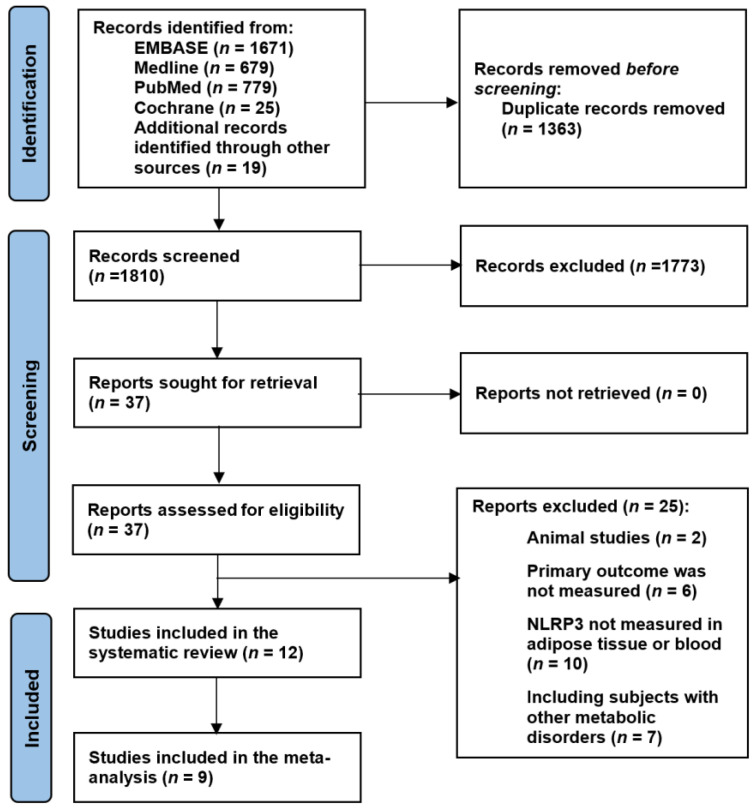
PRISMA flow chart.

**Figure 2 ijms-24-10976-f002:**
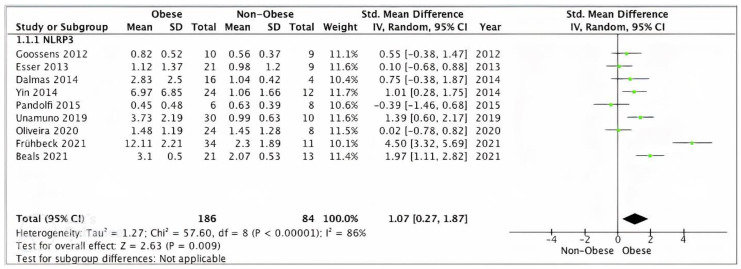
Overall SAT/VAT NLRP3 pooled analysis of 9 studies.

**Figure 3 ijms-24-10976-f003:**
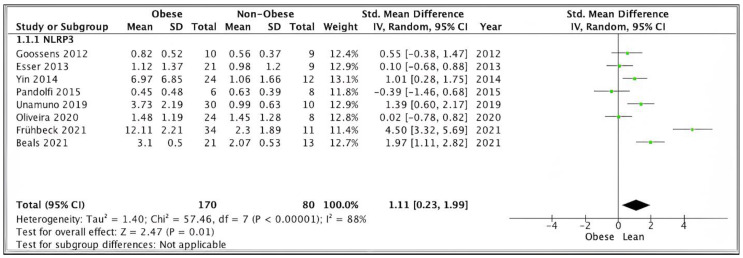
**SAT/VAT** NLRP3 sensitivity pooled analysis of 8 studies.

**Figure 4 ijms-24-10976-f004:**
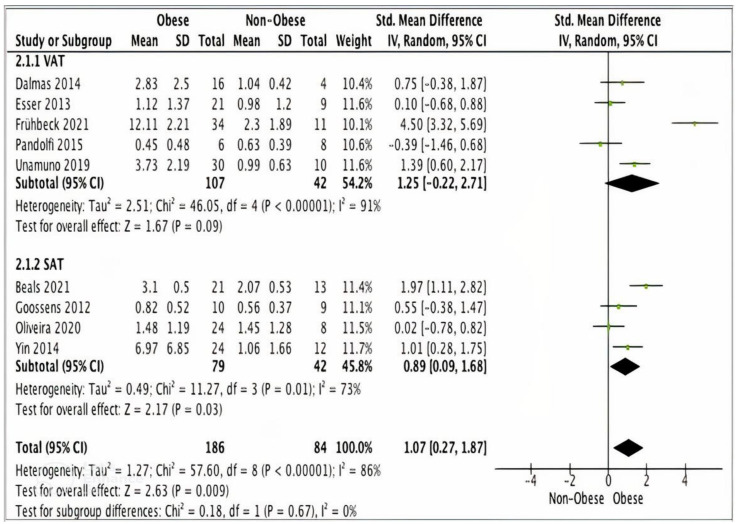
A meta-analysis of NLRP3 from VAT and SAT samples.

**Figure 5 ijms-24-10976-f005:**
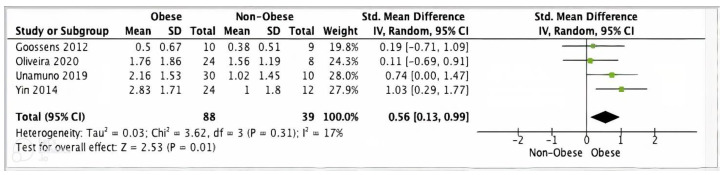
IL-1β meta-analysis of four studies.

**Figure 6 ijms-24-10976-f006:**
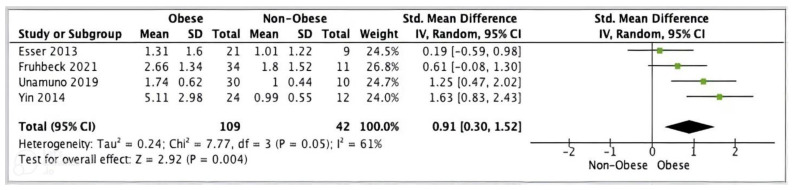
ASC meta-analysis of three studies.

**Table 1 ijms-24-10976-t001:** Modified Newcastle–Ottawa scale for risk of bias and quality assessment of the included studies.

First Author, Year	Selection	Comparability	Outcome	Overall
Beals, 2021 [48]	***	**	**	7
Dalmas, 2014 [45]	***	*	**	6
Esser, 2013 [28]	****	**	**	8
Frühbeck, 2021 [49]	***	**	**	7
Goossens, 2012 [35]	***	**	**	7
Liu, 2018 [47]	**	**	**	6
Oliveira, 2020 [43]	***	**	**	7
Pandolfi, 2015 [46]	***	**	**	7
Serena, 2016 [37]	***	**	**	7
Unamuno, 2021 [50]	***	**	**	7
Williams, 2022 [44]	***	*	**	6
Yin, 2014 [36]	***	**	**	7

Asterisks (*) indicate the star rating for each study with a maximum of four stars for selection, two for comparability, and three for outcome criteria.

## Data Availability

Not applicable.

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
