# Peer review of "The Role of NLRP3 Inflammasome in Obesity and PCOS—A Systematic Review and Meta-Analysis"

_ijms, 2023, doi:10.3390/ijms241310976_

Round 1
Reviewer 1 Report
I would like to congratulate the authors for this interesting and well-written manuscript.
I only have one comment: I think that in the section 3.4.1, where the authors present the findings of studies of NLRP3 expression in human PCOS, they should emphasize that in none of these studies were PCOS subjects and controls matched for BMI. In the study by Rostamtabar et al there are more obese subjects in the PCOS group, in the one by Liu et al the BMI of PCOS subjects is obviously higher than that of controls, and in the study by Guo et al there is no data regarding the participants' BMI. Since obesity is a major determinant of the expression of NLRP3, these results should be interpreted with extreme caution, and this could also be included in the limitations of this review.
Author Response
1. We appreciate your encouraging and positive feedback.
2. We would like to thank the reviewer for this excellent and valid point. We have added a paragraph in the results section to reflect this point. We have also added a comment in the limitations. All changes are highlighted in the revised manuscript:
Line 440-444: It is worth noting that neither of the above three studies matched their PCOS and healthy participants or adjusted their results for BMI (Table 3) [60-62]. Both Rostamtabar et al and Liu et al included more obese subjects in the PCOS groups compared to the controls (Table 3) [61,62); while Gue et al did not report the BMI data for either of the study groups [60].
Line 630 - 634: Furthermore, as mentioned above, none of the human studies matched their study groups or adjusted their results for BMI. As obesity is a major determinant of NLRP3 expression, the results of these studies should be interpreted with extreme caution.
Reviewer 2 Report
Overall the review is well written. Here are my recommendations:
1) The paper is a bit lengthy. The review should focus on human studies only. They should remove the animal studies altogether.
2) The review lacks strength due to a small number of studies. There is nothing much the authors can do about it.
3) Tables: Some data are duplicated in tables. For example, the sample size in Tables 2 and 4. They should remove the sample size columns in Table 4 and state in the footnote "See Table 2 for sample size".
4) All abbreviations should be expanded at first mention in the abstract, text of the paper, table, and figure.
5) There are too many acronyms in the paper. Please unabbreviate AT (adipose tissue).
6) There are too many small paragraphs. Please consolidate these.
Author Response
Comment: Overall the review is well written. Here are my recommendations:
Response: thank you for the positive feedback
Comment 1) The paper is a bit lengthy. The review should focus on human studies only. They should remove the animal studies altogether.
Response: we very much appreciate this valid point. However, the reason for the length of this article is that it includes two systematic reviews. With the scarcity of the human studies on PCOS, we feel that inclusion of the animal studies provides essential preliminary data, which are relevant to the topic. Furthermore, the article is well structured, and it is easy for readers to focus on their section of interest and leave out other parts.
Comment 2): The review lacks strength due to a small number of studies. There is nothing much the authors can do about it.
Response: we fully agree with the reviewer and this point is discussed in detail under the limitations section.
Comment 3) Tables: Some data are duplicated in tables. For example, the sample size in Tables 2 and 4. They should remove the sample size columns in Table 4 and state in the footnote "See Table 2 for sample size".
Response: Thank you for this observation. To avoid duplication, we have removed the sample size columns from table 4 and added a footnote as suggested.
Comment 4) All abbreviations should be expanded at first mention in the abstract, text of the paper, table, and figure.
Response: we have gone through all the abbreviations and spelt them out when first mentioned as highlighted in some places (e.g. IVF)
Comment 5) There are too many acronyms in the paper. Please unabbreviate AT (adipose tissue).
Response: we thank the reviewer and agree with this valuable point. We have removed the “AT” abbreviation and replaced it with “adipose tissue” as highlighted throughout the manuscript including figures and tables (all highlighted).
Comment 6) There are too many small paragraphs. Please consolidate these.
Response: we appreciate this comment and we have carefully reviewed and merged several short paragraphs whenever possible (e.g. the last two paragraphs in the introduction). Please note that many short paragraphs cover separate subheadings and cannot be merged.
Round 2
Reviewer 2 Report
The authors have addressed all my recommendations.